# Functionalized Poly(arylene ether nitrile) Porous Membrane with High Pb(II) Adsorption Performance

**DOI:** 10.3390/polym11091412

**Published:** 2019-08-28

**Authors:** Xiaocan Liu, Meirong Zhou, Xuefei Zhou, Lingling Wang, Xiaobo Liu

**Affiliations:** Research Branch of Advanced Functional Materials, School of Materials and Energy, University of Electronic Science and Technology of China, Chengdu 61173, China

**Keywords:** sulfonated poly(arylene ether nitrile), adsorption, lead ions, cetyltrimethyl ammonium bromide

## Abstract

Porous materials with high specific surface area possess a broad application prospect in the treatment of wastewater. In this work, sulfonated poly(arylene ether nitrile) (SPEN) functionalized with a carboxylic acid group was successfully synthesized, which was subsequently transformed into SPEN porous membranes with cetyltrimethyl ammonium bromide (CTAB) as pore-forming agents to study the adsorption performance for lead ions in aqueous solution. Then, experiments were conducted to investigate the effect of pH, contact time and initial solution concentration on the adsorption performance of porous membranes, and the adsorption capacities of porous membranes with different content (0, 5 and 15 wt %) of CTAB were 183.60, 161.73 and 127.43 mg/g, respectively, which manifested that the adsorption capacity decreased with the increase of CTAB. The adsorption capacities of porous membranes increased with the increase of the initial concentration of lead ions, and the maximum reached was 246.96 mg/g. The simulation of adsorption kinetics revealed that the adsorption was accorded with the pseudo-second-order kinetic model and Langmuir equation, indicating that the adsorption process followed Langmuir monolayer adsorption. Thermogravimetric analysis demonstrated that the porous membranes had excellent thermodynamic properties both before and after adsorption. In addition, the change of adsorption peak in the Attenuated Total Reflectance Fourier Transform Infrared Spectroscopy (ATR-FTIR) spectrum indicated that the absorption performance of porous membranes for lead ions benefited from the chelation between lead ions and the carboxylic acid group on SPEN. Moreover, the porous membranes maintained excellent adsorption properties after circulating five times under the conditions of acidic or alkaline, and the cycle regeneration effect was outstanding.

## 1. Introduction

The rapid development of modern industry brings wealth to society, but it also causes damage to the environment at the same time. Heavy metal ion pollution is particularly serious, which is mainly induced by unscrupulous discharge sewage of industrial effluent without treatment [1]. Industrial waste water contains a variety of harmful heavy metal ions, such as lead ions, cadmium ions and chromium ions, which will exist in the biosphere for a long time and endanger the environment [2]. Particularly, the aforementioned heavy metal ions are harmful to human health. Lead is considered as one of the most toxic heavy metal ions, on account of its wide distribution in the environment and chemical toxicity to various organisms. Low concentrations of lead ions are harmful to the liver, nervous system and kidneys, which can cause abdominal pain, kidney problems, headaches, neurological disorders and even cancer. However, with the application of lead in battery, electroplating, mining, metal processing and other industries, lead ions are constantly produced. Therefore, the problem of wastewater contaminated with lead ions needs to be solved urgently [3,4].

At present, commonly used methods to remove heavy metals include chemical precipitation [5,6], coagulation [7], electrolytic process [8,9], biological method [10,11], ion exchange method [12], adsorption [13,14] and so on. Nevertheless, the methods mentioned above have some drawbacks, such as high cost, high toxicity, large installation space requirements and even bring in secondary pollution [15]. Recently, membrane separation technology has been generally applied in many water treatment fields [16,17]. Membrane technology has received considerable attention because of the advantages of its technical simplicity and environmental compatibility. It is known that a large surface area and adequate adsorption site for membranes are critical for membrane technology in water treatment processes. Therefore, separation membranes with a porous structure and specific functional groups (such as sulfonic acid group, carboxyl group, hydroxyl group, etc.) have attracted the attention of many researchers [18,19]. The separation process is simple and efficient and can be synchronized with other separation and purification processes. Particularly, no additives are required during membrane separation [20]. Moreover, the introduction of a surfactant in the preparation process has been acknowledged to be an efficient method for increasing the porosity of a porous membrane [21,22]. 

Poly(arylene ether nitrile) (PEN), as a special engineering material, possesses excellent thermal stability, corrosion resistance and maneuverability on account of its rigid benzene ring structure and adjustable side chain [23,24]. Meanwhile, it can be prepared into microspheres, fibers, membranes, etc., owing to the diversity of its preparation processes. On the basis of sulfonated poly(arylene ether nitrile) (SPEN) containing sulfonic acid and carboxylic acid groups, the synthesized SPEN membrane exhibits high sensitivity to copper ions (Cu^2+^) and the newly fabricated SPEN absorbent is found to be efficient in the selective adsorption of cationic dyes [25,26]. 

In this work, SPEN containing carboxylic acid groups in the side chains was synthesized, leading to further fabrication of the porous membrane with different amounts of cationic surfactant cetyltrimethyl ammonium bromide (CTAB). In the preparation of the porous membrane, different proportions of CTAB were added to study the effect of cationic surfactant on the adsorption of lead ions in an aqueous solution. Then the porous membrane with the largest adsorption capacity for lead ions was systematically studied in combination with isothermal adsorption. Furthermore, we also studied the influence of pH, contact time and initial concentration on the adsorption capacity for Pb(II) and adsorption mechanism of the porous membrane for Pb(II). Compared with other works, we used a new polymer to fabricate the porous membrane, and then the ability of porous membranes to remove lead ions under various conditions was studied experimentally. Moreover, the adsorption capacity for lead ions by porous membranes was higher than that of other literature [27,28,29,30,31,32,33].

## 2. Materials and Methods 

### 2.1. Materials

Hydroquinone sulfonic acid potassium salt (SHQ) and 2,6-difluorobenzonitrile (DFBN) were purchased from Shanghai Sigma Aldrich (Shanghai, China). Phenolphthalein (PP), sodium hydroxide (NaOH, AR), zinc (Zn), *N*-methyl pyrrolidone (NMP, AR), potassium carbonate (K_2_CO_3_, AR), ethanol, toluene, cetyltrimethyl ammonium bromide (CTAB) and lead nitrate (PbNO_3_) were supplied by Chengdu Kelong Chemical Reagent Company (Chengdu, China). 

### 2.2. SPEN and Porous Membrane Preparation

Firstly, phenolphthalein (PPL) was synthesized from PP, NaOH and Zn [34]. Then, PPL (22.4 g, 0.07 mol), SHQ (6.84 g, 0.03 mol), DFBN (13.9 g, 0.1 mol), K_2_CO_3_ (28 g), NMP (65 mL) and toluene (25 mL) were added to a three-necked flask. Among them, PPL, SHQ and DFBN were the raw materials for the synthesis of SPEN, NMP was the solvent, K_2_CO_3_ acted as a catalyst to facilitate polymerization and toluene was the dehydrating agent. They were evenly mixed and then heated up to 145 °C and maintained for 3 h in the stage of dehydration. Then, the temperature was slowly raised to 180 °C and kept for 1 h. At the end of the polymerization, the product was precipitated in ethanol. Then, the product was sequentially washed with ethanol, hydrochloric acid and deionized water to remove the unreacted raw materials, such as PPL, SHQ, DFBN and K_2_CO_3_. Finally, the product was dried in a vacuum oven at 80 °C [35].

First, the synthesized SPEN accompanied by 0, 5 and 15 wt % CTAB were dissolved in NMP and coated on a glass plate by a 200 μm applicator, respectively. NMP was the solvent, and CTAB was a kind of hydrophilic surfactant, which can reduce the interfacial tension between the membrane and water, so as to promote the growth of pores. This was the dissolution process, in which there was no phase change. Second, the liquid film was immersed into a coagulation bath (HCl solution). Then, due to the diffusion exchange of the solvent (NMP) and nonsolvent (HCl solution), the solubility of polymer in the system decreased and thermodynamics changed, and phase separation occurred. Finally, the porous membrane was soaked in a coagulation bath for 24 h. After the obtained porous membranes were immersed in 1 mol/L NaOH solution for 2 h, they were washed with deionized water to remove the residual NaOH and then dried in an oven for the next applications. The resulting porous membranes with different CTAB content were named as SPEN-CTAB-0%, SPEN-CTAB-5% and SPEN-CTAB-15%.

### 2.3. Adsorption Behaviour for Pb(II)

The effect of pH (3, 4, 5, 6, 7), CTAB content (0, 5, 15 wt %), contact time (0, 5, 15, 30, 60, 90, 120 min) and initial concentration (50, 70, 100, 150, 200 mg/L) to the adsorption behavior of the porous membrane for Pb(II) were investigated. In a typical experiment, 25 mg SPEN-CTAB-0% was immersed in 200 mg/L PbNO_3_ solution, the pH values were regulated by a solution of 0.1 mol/L HCl and 0.1 mol/L NaOH. Then, adsorption experiments were carried out using three different CTAB content porous membranes (25 mg) in 150 mL of a 70 mg/L PbNO_3_ solution. Functionalized adsorption for Pb(II) from aqueous solution was measured by kinetics similar to adsorption isotherms. The concentration of Pb(II) was measured by inductively coupled plasma optical emission spectrometer (ICP-OES). Moreover, 0.1 mol/L HCl solution, 0.2 mol/L NaOH solution and deionized water were used as regeneration agents for the adsorption–desorption cycle experiment, respectively. All adsorption experiments were carried out at 298.15 K with the pH at 5. The instantaneous adsorption amount (*q*_t_) and equilibrium adsorption amount (*q*_e_) of Pb(II) were calculated by Equations (1) and (2) [26]:
(1)$$q_{t}=\left(\frac{C_{0}-C_{t}}{m}\right)\times V$$
(2)$$q_{e}=\left(\frac{C_{0}-C_{e}}{m}\right)\times V$$
where *C*_0_ (mg/L), *C*_t_ (mg/L) and *C*_e_ (mg/L) are the initial concentration of Pb(II), the residual concentration at time *t* (min) and the residual concentration at the time of adsorption equilibrium, respectively. *V* (mL) is the volume of the Pb(II) solution, and *m* (mg) is the mass of the porous membrane.

### 2.4. Characterization

The structure and composition of SPEN were determined by Bruker AV II-400 spectrometer and Fourier transform infrared spectroscopy (FTIR and ATR-FTIR) instrument (Shimadzu 8400S, Kyoto, Japan). Among them, the Bruker AV II-400 spectrometer measured the ^1^H NMR (400 MHz) spectra relative to DMSO-d6 (H:d = 2.50 ppm) as a reference to measure the SPEN nuclear magnetic spectrum. Thermal stability of SPEN was determined by Thermogravimetric Analysis TGA-Q50 (TA Instruments, Newcastle, DE, USA). The weight-average molecular weight (*M*_w_), number average molecular weight (*M*_n_) and polydispersity index (PDI; *M*_w_/*M*_n_) of SPEN were measured by gel permeation chromatography (GPC) with polystyrene as the standard sample by PL-GPC220 system. The cross-section and surface microstructure of the three porous membranes were characterized by scanning electron microscopy (SEM) (JSM-6490LV, JEOL, Akishima, Japan). The hydrophilicity of the porous membrane was characterized by measuring the contact angle of the porous membrane surface at room temperature. The equilibrium water content (EWC) and porosity of the porous membrane were calculated from Equations (3) and (4) by measuring the water absorption (298.15 K, 24 h) of the porous membrane in deionized water [36]:
(3)$$\mathrm{EWC}\;\left(\%\right)=\frac{W_{w}-W_{d}}{W_{w}}\times 100$$
(4)$$\mathrm{Porosity}=\frac{W_{w}-W_{d}}{\rho_{w}\times V}$$
where *W*_w_ (g) and *W*_d_ (g) are the masses of dry membrane and wet membrane, respectively. *ρ*_w_ (g/cm^3^) is the density of water, and *V* (cm^3^) is the volume of the wet membrane. Moreover, the porosities of different membranes were tested by an automatic mercury injection apparatus (Micromeritics Instrument Corporation, IV9500, Norcross, GA, USA). The Pb(II) concentration was tested by ICP-OES (Peakinelmer Optima 7300, Waltham, MA, USA).

## 3. Results and Discussion 

### 3.1. Characterization of SPEN

The synthesis process of SPEN via nucleophilic substitution reaction is displayed in Figure 1a and the infrared spectrum of raw SPEN is shown in Figure 1b. It was clear that the absorption peak at 2230 cm^−1^ was ascribed to the stretching bond of –CN, the peak at 1701 cm^−1^ was the stretching of –COOH, and the peaks at 1576 and 1450 cm^−1^ were attributed to the stretching vibration of the benzene ring. The ether bond absorption peak was observed at 1245 cm^−1^ and sulfonic acid absorption peak was located at 1075–1024 cm^−1^ [26,34]. From the nuclear magnetic resonance spectrum in Figure 1d, the peak at 12.98 ppm was the characteristic peak of H on –COOH, the characteristic peaks of H on the benzene ring were located around 6.39–7.80 ppm, characteristic peaks of C–H varied from 1.04 to 2.68 ppm. Figure 1c is the TGA curves of raw SPEN and 5% weight loss temperature was 430 °C, which revealed that the prepared SPEN possessed excellent thermal stability. *M*_w_, *M*_n_ and PDI of SPEN measured by GPC were 59,342, 69,221 g/mol and 1.166, respectively.

### 3.2. Structure Characterization of Porous Membrane

The cross-section and surface morphology of the porous membranes are demonstrated in Figure 2. Figure 2a–c show the cross-sectional morphology of three different porous membranes with a scale of 40 μm, respectively. Figure 2a1–c1 show the cross-sectional morphology of different porous membranes at 10 μm. It can be seen that the porous membranes consist of a dense top layer and porous supporting layer, and the porous supporting layer is the subject part of the porous membrane. From Figure 2a–c, the pore size in the porous supporting layer of SPEN-CTAB-15% was significantly increased compared to SPEN-CTAB-0% and SPEN-CTAB-5%. Therefore, it can be obtained that the pore diameter in the porous supporting layer of the porous membrane accordingly increased as the increase of the proportion of surfactant CTAB. However, with the increases of CTAB, the pores of the dense top layer decreased and became denser by comparing the surface morphologies of different porous membranes (Figure 2a–c2).

The water absorption of porous membranes containing different proportions of CTAB is shown in Figure 3. The equilibrium water content and porosity is displayed in Table 1. It can be seen from Figure 3 that both the water absorption rate and surface contact angle of the porous membrane increased with the increase in additional CTAB content. The water absorption rates of SPEN-CTAB-0%, SPEN-CTAB-5% and SPEN-CTAB-15% were 283, 285 and 385 wt %, respectively. Furthermore, the contact angles were 54.7°, 61.9° and 70.6°, respectively. The EWC and porosity in Table 1 also increased as CTAB increased, and the highest porosity reached 83.60%. Moreover, the porosities of SPEN-CTAB-0%, SPEN-CTAB-5% and SPEN-CTAB-15% which were tested by an automatic mercury injection apparatus were 71.98%, 75.53% and 83.14%, respectively. The result was consistent with the porosity calculated by Equation (4).

With the augment of CTAB, the equilibrium water content and porosity accordingly increased, which conformed to the increase of pore size in the porous supporting layer (Figure 2 and Figure 3). Furthermore, the increase in CTAB concentration assisted the porous membrane with a denser surface morphology, which also contributed to the increase of the contact angle.

### 3.3. Effect of Solution pH

The effect of solution pH on the adsorption capacity for Pb(II) using SPEN-CTAB-0% under 200 mg/L Pb(II) aqueous solution and a contact time of 2 h (298.15 K) is shown in Figure 4. It can be concluded from Figure 4 that the adsorption capacity was distinctly affected by pH values. The adsorption capacity increased with the pH increasing from 3 to 6, while the adsorption capacity decreased with the continuous increase of pH values from 6 to 7. The results may be induced by the competition between H_3_O^+^ and Pb(II) at pH 3 and 4, the protonation of –COOH and –SO_3_H groups under strong acidic conditions which resulted in low adsorption capacity [27], and the formation of hydroxyl complexes hinders the chelation between Pb(II) and a functional group at a high pH value [31]. The maximum adsorption capacity was up to 313.81 mg/g at pH 6. However, the spontaneous precipitation of Pb(II) would come up and generate Pb(OH)_2_ sediment when the pH value was 6 and higher. Therefore, after this experiment pH values were set as 5, which is also a familiar pH value in industrial wastewater [30]. Although the adsorption capacity would be reduced when the optimal pH value was not reached, the adsorption capacity was still considerable in general. Moreover, the experimental results would not be disturbed by the hydroxyl complexes of Pb(II), which was relatively reliable.

### 3.4. Effect of CTAB Content in a Porous Membrane

The effects of contact time on the adsorption of porous membranes with different CTAB ratios were studied at definite time intervals ranging from 0 to 120 min. All adsorption experiments were carried out by placing 25 mg of SPEN porous membranes into 150 mL of 70 mg/L PbNO_3_ solution. As shown in Figure 5a, the adsorption capacity increased rapidly in the first 30 min and reached an adsorption equilibrium in 1 h. The initial rapid adsorption of Pb(II) indicated a high affinity between Pb(II) and the porous membranes, which may be attributed to the large surface area of the SPEN porous membranes that enhance the exposure of functional groups. The equilibrium adsorption capacities of SPEN-CTAB-0%, SPEN-CTAB-5% and SPEN-CTAB-15% were 183.6, 161.4 and 127.4 mg/g, respectively. In order to better understand the adsorption behavior, the adsorption data were analyzed on the basis of two commonly used kinetic models, namely the first-order kinetic model and second-order kinetic model shown in Equations (5) and (6), respectively [23,26]:
(5)$$log(q_{e}-q_{t})=logq_{e}-\frac{k_{1}}{2.303}t$$
(6)$$\frac{t}{q_{t}}=\frac{1}{k_{2}q_{e}^{2}}+\frac{t}{q_{e}}$$
where *t* (min) is the adsorption time, *k*_1_ (min^−1^) is the rate constant of pseudo-first-order and *k*_2_ (g/mg min) is the rate constant of pseudo-second-order.

Figure 5b,c show the pseudo-first-order kinetics model fitting and the pseudo-second-order kinetic model fitting of the porous membrane for Pb(II), and the relevant parameters of the kinetic fitting are listed in Table 2. From Figure 5b,c and the correlation coefficients in Table 2, it can be clearly seen that the adsorption kinetics of Pb(II) by the three porous membranes are all consistent with the second-order kinetic model. The *R*^2^ of SPEN-CTAB-0%, SPEN-CTAB-5% and SPEN-CTAB-15% calculated by the pseudo-second-order kinetic model were 0.9911, 0.9693 and 0.9978, respectively. Furthermore, the theoretical adsorption capacity obtained by pseudo-second-order kinetic model was close to the experimentally measured adsorption capacity. Therefore, the adsorption process of Pb(II) by the porous membrane followed the pseudo-second-order model, which indicated that the adsorption process was rapid and probably dominated by chemical adsorption [26,37].

From the data, as the ratio of CTAB increased, the adsorption capacity of the porous membrane for Pb(II) decreased. This was probably because the cationic surfactant CTAB consumed certain adsorption sites of a functional group (–COO) on the SPEN membrane for further adsorption. In addition, micelles may have accumulated on the membrane surface and caused membrane contamination [38,39]. The steps of the adsorption process are generally studied by an intermolecular diffusion model, as in Equation (7) [26]:
(7)$$q_{t}=k_{i}t^{0.5}+C$$
where *k_i_* is the diffusion rate constant and *C* is the intercept that is influenced by the thickness of the boundary layer. Figure 5d is an intermolecular diffusion model fitting and the parameters are listed in Table 2. It can be seen from the fitted graph that the whole adsorption process for Pb(II) can be divided into two stages: firstly adsorbing on the surface and then slowly diffusing into the pores of the membrane. Moreover, the two curves did not pass the origin, indicating that the internal diffusion was not the only speed control step in the entire adsorption process. 

### 3.5. Effect of Initial Concentrations

Next, the PbNO_3_ solutions with concentrations ranging from 50 to 200 mg/L were prepared, and the adsorption capacity and adsorption kinetics of SPEN-CTAB-0% to Pb(II) at different initial concentrations were simulated. From Figure 6a, the adsorption reached equilibrium in half an hour, and the adsorption rate was fast in the whole adsorption process. In addition, the maximum adsorption capacities of the porous membrane continuously increased along with the increase in initial concentrations. 

The equilibrium adsorption amounts of SPEN-CTAB-0% in the presence of Pb(II) solutions with the concentrations from 50 to 200 mg/L were 195.82, 196.09 202.99, 217.92 and 246.96 mg/g, respectively, which were outstanding and larger than those of reported adsorbents for Pb(II) removal by other researchers as demonstrated in Table 3. The corresponding results of adsorption kinetics (Figure 6b,c) and the intermolecular diffusion model (Figure 6d) for Pb(II) are listed in Table 4. The *R*^2^ of the pseudo-second-order kinetic model are 0.9896, 0.9975, 0.9978, 0.9990 and 0.9951, respectively. The maximum adsorption capacity obtained by pseudo-second-order kinetics model was close to the experimental value. Therefore, the adsorption process mainly followed the chemisorption mechanism and involved electron sharing or electron transfer between the adsorbent and adsorbate. It can be seen from Figure 6d that the whole adsorption process was also divided into two stages, and the fitting curve did not pass through the origin, so the intermolecular process was not the rate limiting step in the adsorption process.

### 3.6. Adsorption Isotherm 

On the basis of equilibrium adsorption capacities (298.15 K, pH = 5) of SPEN-CTAB-0% to Pb(II) in different concentrations, Langmuir (Equation (8)) and Freundlich (Equation (9)) models were used to analyze the adsorption isotherm [1]:
(8)$$\frac{C_{e}}{q_{t}}=\frac{1}{K_{L}q_{m}}+\frac{C_{e}}{q_{m}}$$
(9)$$ln\;q_{e}=ln\;K_{F}+\frac{1}{n}ln\;C_{e}$$
where *K*_L_ (L/mg) is the Langmuir adsorption equilibrium constant and *q*_m_ (mg/g) is the maximum adsorption capacity. *K*_F_ ((mg/g) (L/mg)^1/*n*^) and 1/*n* are Freundlich constants. The Langmuir model assumes that the adsorbent surface is uniform and there is no interaction among adsorbents. The adsorption process is a single layer adsorption, that is, adsorption only occurred on the surface of the adsorbent. The Freundlich model can be applied to both single-layer adsorption and non-uniform surface adsorption, which can describe the mechanism of uneven surface adsorption well [23,26]. According to Figure 7 and Table 5, the obtained *R*^2^ of the Langmuir model (0.9807) was larger than that of the Freundlich model (0.8046), which indicated that the Langmuir model was more applicable to analyze the adsorption isotherm. Moreover, the maximum adsorption capacity (281.69 mg/g) simulated by the Langmuir model approached the *q*_m_ (246.96 mg/g) obtained in the experiment. Therefore, the adsorption process of porous membranes to Pb(II) was monolayer adsorption. The adsorption occurred when the adsorbate collided with the empty adsorption sites, and one adsorbent occupied one adsorption site. In addition, the separation factor *R_L_* calculated by the Langmuir model can be used to evaluate the feasibility of the adsorption process, as in Equation (10):
(10)$$R_{L}=\frac{1}{1+K_{L}C_{0}}$$


When *R*_L_ = 0, the isotherm is irreversible, *R*_L_ = 0~1 is favorable, the isotherm is linear when *R*_L_ = 1, and *R*_L_ > 1 is unfavorable [26]. The *R*_L_ calculated from five different initial concentrations of Pb(II) were 0.1443, 0.1210, 0.08426, 0.05783 and 0.04250, respectively. *R*_L_ was less than 1, which demonstrated that SPEN-CTAB-0% was favorable for the adsorption of Pb(II).

### 3.7. Thermal Stability Changes before and after Adsorption

As can be seen from Figure 8, the TGA curves of membranes changed after adsorption compared to the membranes before adsorption, which may be caused by the combination of lead ions and groups in SPEN porous membranes. The 5% thermogravimetric temperature of porous membranes before adsorption was 430 °C, and the 5% thermogravimetric temperature had little difference after adsorption in Pb(II) solutions of porous membranes at different concentrations, which was about 400 °C. At 600 °C, the mass weight loss of the membrane before adsorption was 26% and 20% after adsorption. The results manifested the exceptional thermal stability of porous membranes.

### 3.8. Adsorption Mechanism

The SPEN-CTAB-0% membrane was subjected to ATR-FTIR test (298.15 K) to gain an in-depth understanding of the adsorption reaction mechanism. The ATR-FTIR spectra of SPEN-CTAB-0% membrane before and after adsorption of Pb(II) with a concentration of 200 mg/L are exhibited in Figure 9a,b. Moreover, the membrane after adsorption was treated using 0.2 mol/L NaOH solution to desorb Pb(II), as shown in Figure 9c.

It could be seen that the carboxylic acid absorption peaks at 1700–1650 cm^−1^ changed due to the carboxylic acid that converted into a carboxylate form after the membrane was treated with the solution of NaOH. Meanwhile, the peaks at 1554 and 1399 cm^−1^ in Figure 9a can be classified as carboxylate adsorption peaks [40]. After Pb(II) adsorption experiment (Figure 9b), an absorption peak appeared at 1711 cm^−1^, the absorption peak at 1554 cm^−1^ disappeared, and the absorption peak at 1399 cm^−1^ was red-shifted (from 1399 to 1387 cm^−1^) and the intensity decreased. After desorption (Figure 9c), an absorption peak appeared at 1551 cm^−1^, and the absorption peak at 1387 cm^−1^ returned to 1400 cm^−1^, which was similar to the peak distribution of the membrane before adsorption. In addition, the absorption peaks at other locations did not change significantly. Therefore, the adsorption for Pb(II) by the SPEN porous membrane may occur through the complexation of Pb(II) with a carboxylic acid group.

### 3.9. Recycling of SPEN Porous Membranes

In order to ensure the objective applicability of the adsorbent, a desorption–adsorption experiment (298.15 K, pH = 5) was performed on the SPEN-CTAB-0% membrane which achieved adsorption equilibrium in 150 mL of 200 mg/L PbNO_3_ solution. The desorption experiment was carried out using 150 mL of 0.1 mol/L HCl solution, 0.2 mol/L NaOH solution and deionized water as a strippant, respectively. After the membrane was desorbed, it was placed in 150 mL of a 200 mg/L PbNO_3_ solution for re-adsorption experiments. For the purpose of obtaining a larger exchange capacity, the adsorption process was conducted by using a porous membrane soaked in NaOH solution. As a control experiment, the porous membrane with HCl as a desorption agent was also soaked in 1 mol/L NaOH solution and dried for re-adsorption. The reusability of the desorbed SPEN-CTAB-0% membrane was examined, as shown in Figure 10. It can be seen that the adsorption capacity of SPEN-CTAB-0% membrane with HCl and NaOH as desorption agents maintained relatively stable after five cycles. The adsorption capacity decreased from 223 to 183 mg/g after five cycles with the assistance of HCl as desorption agent, and the adsorption capacity decreased from 224 to 190 mg/g after five cycles using NaOH as desorption agent. However, the adsorption capacity decreased from 204 to 4 mg/g after five cycles when water acted as desorption agent. The decreased adsorption capacity probably ascribed to the covalent bond of water which was too weak and could not destroy the complexation of –COO and Pb(II). Therefore, the adsorption sites on the regenerated membrane were occupied by the initially adsorbed Pb(II), leading to poor cyclic adsorption performance. Furthermore, the desorption rate of SPEN-CTAB-0% using water as a strippant at different time intervals is listed in Figure 11. The desorption rate was calculated by Equation (11):
(11)$$\text{Desorption rate}=\frac{C_{2}}{C_{0}-C_{1}}\times 100\%$$
where *C*_2_ is the concentration after desorption, *C*_0_ is the concentration of the initial solution and *C*_1_ is the concentration after adsorption. Figure 11 displays the Pb(II) desorption rate of the porous membrane with deionized water as the desorbent (298.15 K, pH = 5) at different times. It exhibited that only 3.05% of Pb(II) was desorbed from the porous membrane, indicating that there were few physically adsorbed Pb(II). In other words, the chemical adsorption dominated the main place in the adsorption process of Pb(II) based on porous SPEN membrane, which was consistent with the results concluded from adsorption kinetics in previous work.

## 4. Conclusions

In summary, the fabricated carboxylic acid functionalized SPEN porous membrane had a competitive adsorption capacity for Pb(II) in aqueous solution, whose adsorption capacity reached up to 246.96 mg/g. We studied the effect of surfactant CTAB on the adsorption capacity. Experimental results showed that the addition of CTAB decreased the adsorption capacity of porous membranes for Pb(II). The reduced adsorption capacity was ascribed to the amount of adsorption sites on the porous membrane which were occupied by CTAB, and the membrane may have been polluted by micelles to some extent. Furthermore, the pseudo-second-order kinetics of the adsorption of porous membranes for Pb(II) was obtained via kinetic simulation, which was consistent with the Langmuir single-layer adsorption model. The change of adsorption peak in ATR-FTIR spectrum indicated that the adsorption of porous membrane for Pb(II) largely profited from the carboxylic acid groups of SPEN. Moreover, the porous membranes possessed good thermal stability and high adsorption capacity after five adsorption-desorption cycles, which were up to 183 and 190 mg/g with HCl and NaOH as regeneration agents. On account of the facile preparation method, great thermal stability, high Pb(II) adsorption capacity and good cycles efficiency for adsorption, the SPEN porous membranes are expected to be applicable in the removal of Pb(II) in wastewater.

## Figures and Tables

**Figure 1 polymers-11-01412-f001:**
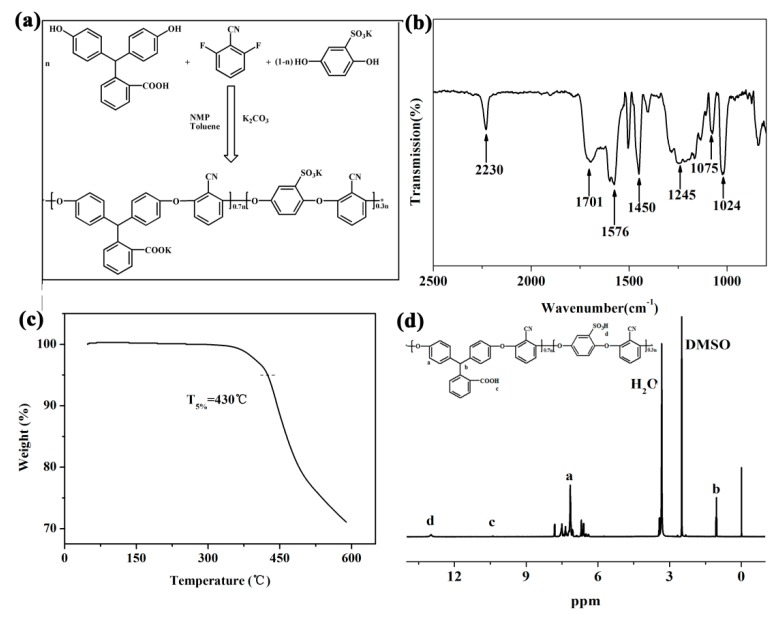
Synthesis route (**a**) FTIR (**b**), TGA curves and (**c**) ^1^H NMR spectra (**d**) of raw sulfonated poly(arylene ether nitrile) (SPEN).

**Figure 2 polymers-11-01412-f002:**
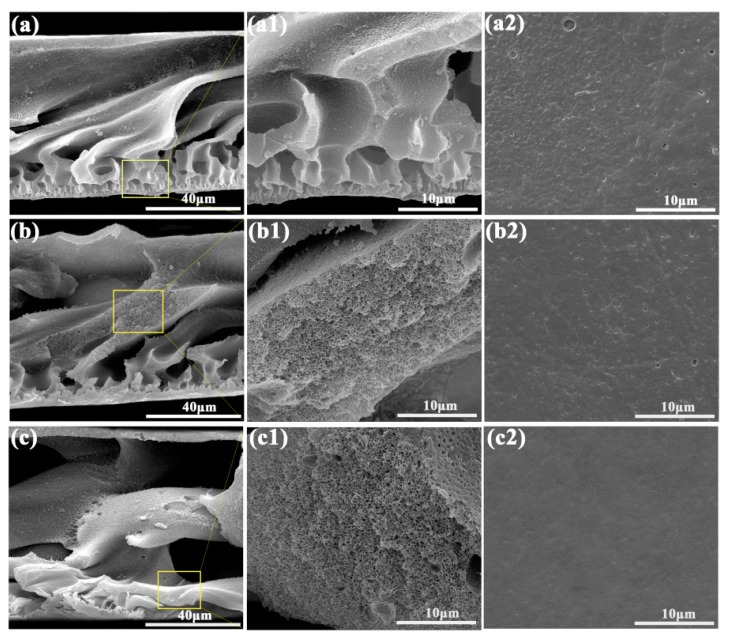
Cross-section SEM images of SPEN-CTAB-0% (**a**), (**a1**); SPEN-CTAB-5% (**b**), (**b1**); SPEN-CTAB-15% (**c**), (**c1**). Surface SEM images of SPEN-CTAB-0% (**a2**), SPEN-CTAB-5% (**b2**) and SPEN-CTAB-15% (**c2**). CTAB: cetyltrimethyl ammonium bromide.

**Figure 3 polymers-11-01412-f003:**
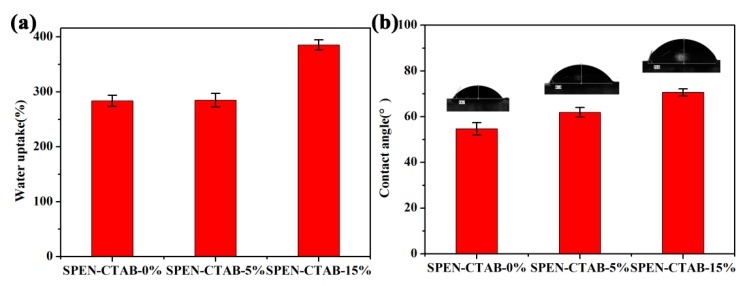
The water uptake (**a**) and contact angle (**b**) of SPEN-CTAB-0%, SPEN-CTAB-5% and SPEN-CTAB-15%.

**Figure 4 polymers-11-01412-f004:**
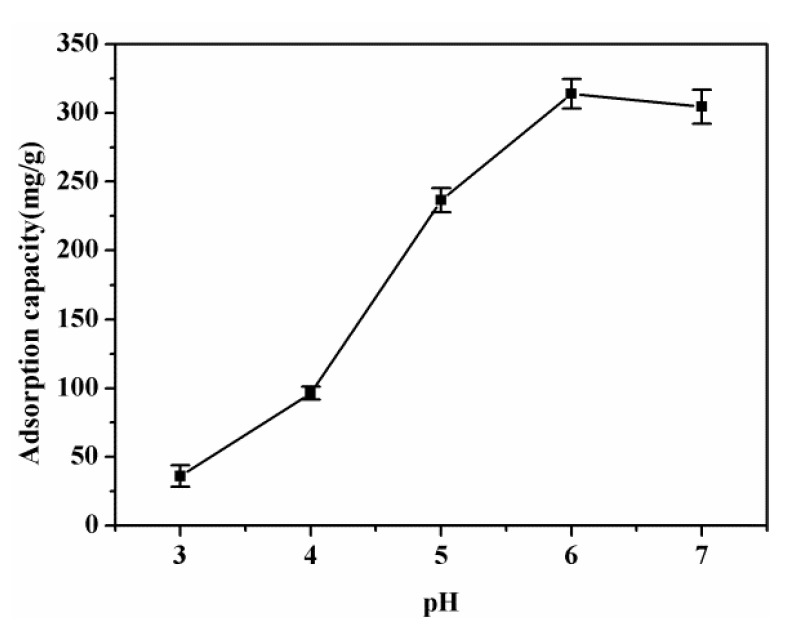
Effect of pH on the adsorption capacity using SPEN-CTAB-0%.

**Figure 5 polymers-11-01412-f005:**
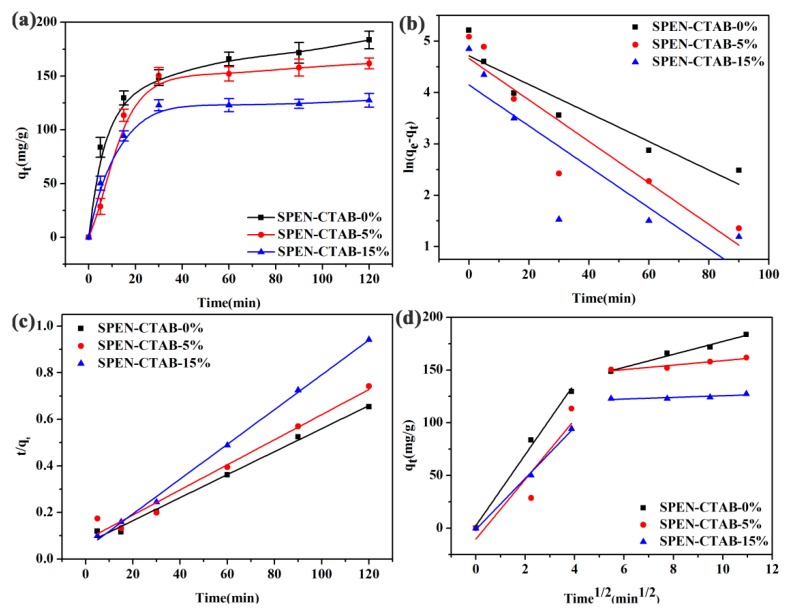
(**a**) Effect of contact time on Pb(II) adsorption capacity, (**b**) pseudo-first-order, (**c**) pseudo-second-order, and (**d**) intraparticle diffusion kinetic models (**d**) for the adsorption of Pb(II) onto SPEN-CTAB-0%, SPEN-CTAB-5% and SPEN-CTAB-15%, respectively.

**Figure 6 polymers-11-01412-f006:**
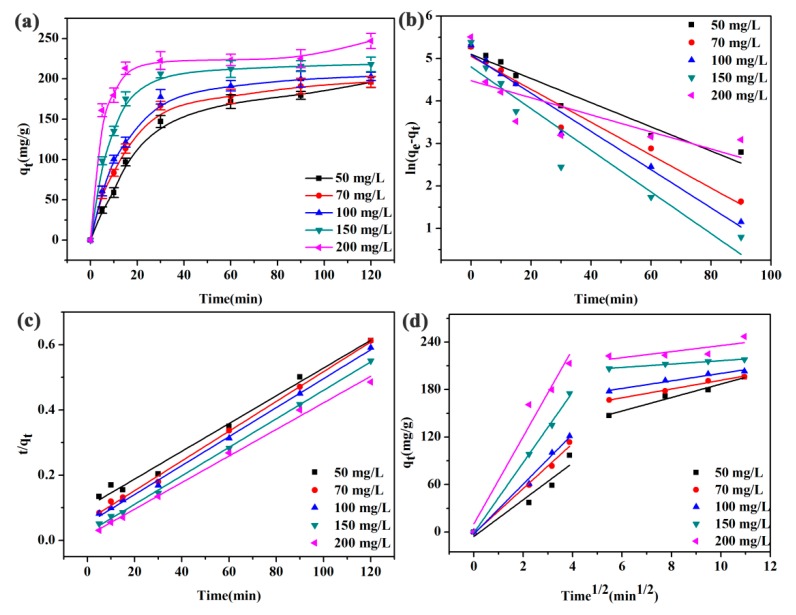
(**a**) Effect of initial concentrations on Pb(II), (**b**) pseudo-first-order, (**c**) pseudo-second-order, and (**d**) and intraparticle diffusion for the adsorption of Pb(II) onto SPEN-CTAB-0%.

**Figure 7 polymers-11-01412-f007:**
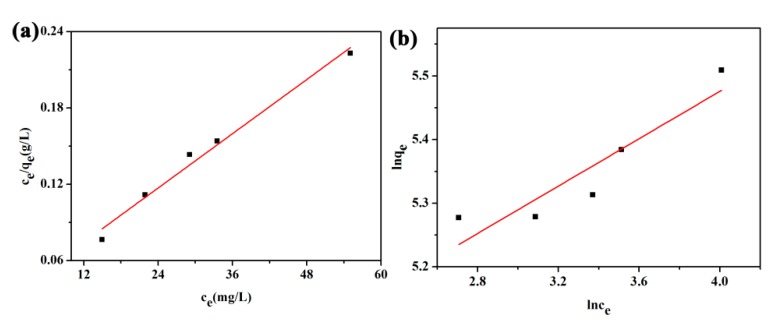
(**a**) Langmuir isotherm and (**b**) and Freundlich isotherm for adsorption of Pb(II) onto SPEN-CTAB-0%.

**Figure 8 polymers-11-01412-f008:**
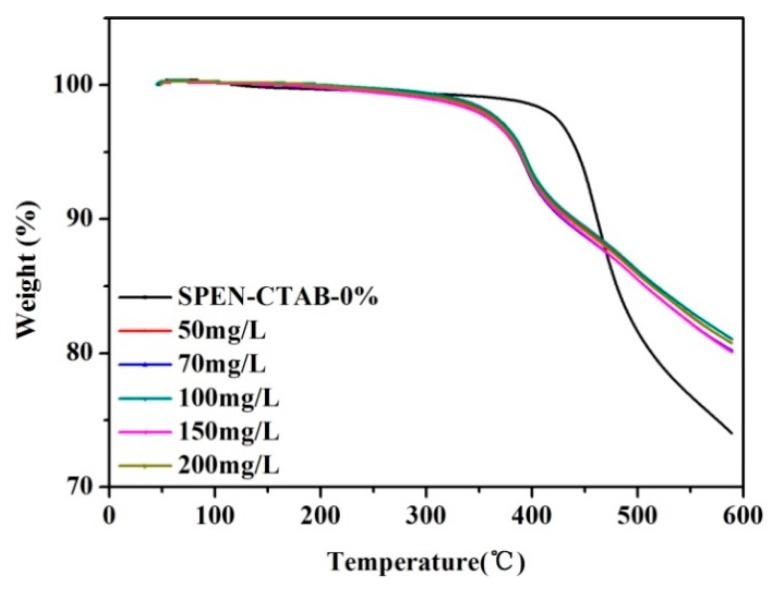
TGA curves of the SPEN-CTAB-0% before and after adsorption.

**Figure 9 polymers-11-01412-f009:**
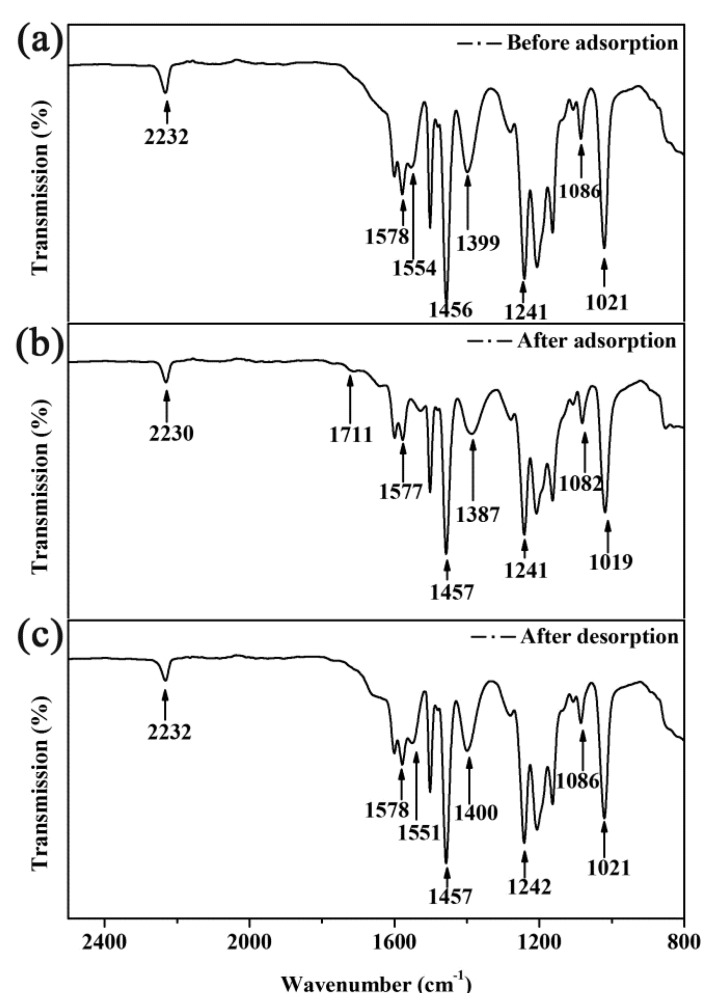
ATR-FTIR of SPEN-CTAB-0% before and after adsorption and desorption.

**Figure 10 polymers-11-01412-f010:**
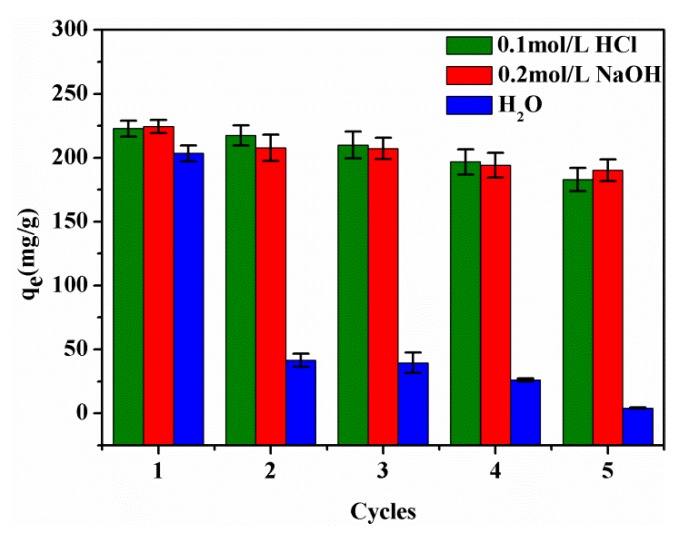
Regeneration of SPEN-CTAB-0% under a different strippant.

**Figure 11 polymers-11-01412-f011:**
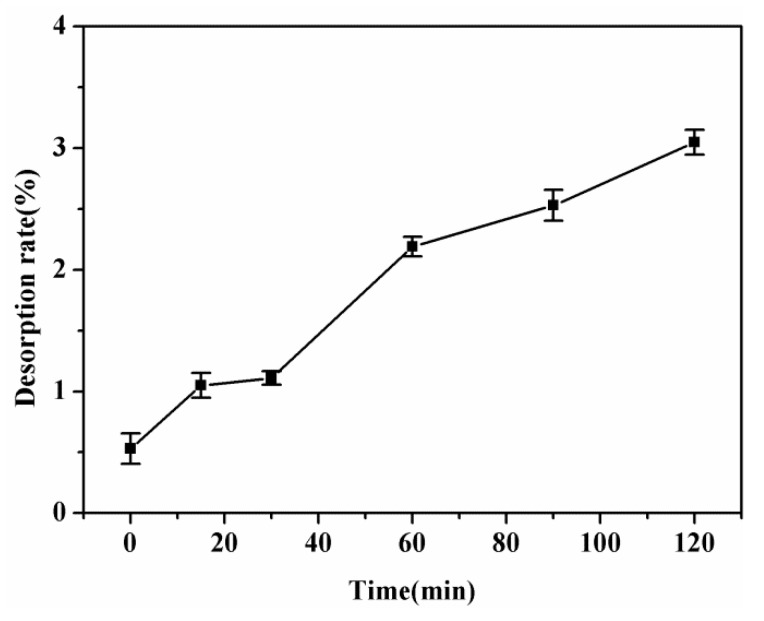
Desorption rate of SPEN-CTAB-0% at different times using water as a strippant.

**Table 1 polymers-11-01412-t001:** The equilibrium water content (EWC), porosity obtained from Equation (4) and contact angle of SPEN-CTAB-0%, SPEN-CTAB-5% and SPEN-CTAB-15%.

Membrane	EWC (%)	Porosity (%)	Contact Angle (°)
SPEN-CTAB-0%	73.93	78.95	54.7
SPEN-CTAB-5%	74.02	79.03	61.9
SPEN-CTAB-15%	79.40	83.60	70.6

**Table 2 polymers-11-01412-t002:** The kinetic parameters of adsorption of Pb(II) onto SPEN-CTAB-0%, SPEN-CTAB-5% and SPEN-CTAB-15%.

Membrane	Parameters	SPEN-CTAB-0%	SPEN-CTAB-5%	SPEN-CTAB-15%
Pseudo-first-order	*k*_1_ (min^−1^)	0.0278	0.0404	0.0398
	*q*_e_ (cal.) (mg/g)	12.811	12.678	11.274
	*q*_e_ (exp.) (mg/g)	183.60	161.73	127.43
	*R* ^2^	0.8703	0.8320	0.6967
Pseudo-second-order	*k*_2_ (g/mg·min)	0.00037	0.00036	0.0012
	*q*_e_ (cal.) (mg/g)	202.43	185.53	134.05
	*q*_e_ (exp.) (mg/g)	183.60	161.73	127.43
	*R* ^2^	0.9911	0.9693	0.9978
Intraparticle diffusion	*k* _*i*1_	33.697	28.316	24.236
	*C*	2.4810	−10.281	−1.170
	*R* _1_ ^2^	0.9879	0.7428	0.9948
	*k* _*i*2_	6.1267	2.1267	0.7810
	*C*	115.84	137.61	117.76
	*R* _2_ ^2^	0.9682	0.8814	0.5995

**Table 3 polymers-11-01412-t003:** Adsorption capacities of Pb(II) on various adsorbents.

Adsorbent	pH	Pb(II) Adsorption Capacity (mg/g)	Reference
PSF/HFO NPs MMMs	6.5–7	13.2	[27]
PES/PDA-R UFAMs	5.4	20.23	[28]
PES/chitosan/PVA/A-Fe_3_O_4_ MMMs	6	40	[29]
PES/HHZO nanoparticles	5	104	[30]
PAN/chitosan/UiO-66-NH_2_ MOF nanofibers	6	115	[31]
PES/PANI@GO MMMs	6	145	[32]
PES/HMO MMMs	7–8	204.1	[33]
SPEN porous membranes	5	246.96	This study

**Table 4 polymers-11-01412-t004:** The kinetic parameters for the adsorption of Pb(II) onto SPEN-CTAB-0%.

*C*_0_ (mg/L)	Parameters	50	70	100	150	200
Pseudo-first-order	*k*_1_ (min^−1^)	0.0284	0.0387	0.0450	0.0490	0.0201
	*q*_e_ (cal.) (mg/g)	13.861	13.730	13.821	13.052	12.174
	*q*_e_ (exp.) (mg/g)	195.82	196.09	202.99	217.92	246.96
	*R* ^2^	0.9370	0.9588	0.9712	0.8994	0.4650
Pseudo-second-order	*k*_2_ (g/mg·min)	1.8 × 10^−4^	3.4 × 10^−4^	3.8 × 10^−4^	8.1 × 10^−4^	1.1 × 10^−3^
	*q*_e_ (cal.) (mg/g)	234.74	219.30	225.22	228.83	245.70
	*q*_e_ (exp.) (mg/g)	195.82	196.09	202.99	217.92	246.96
	*R* ^2^	0.9896	0.9975	0.9978	0.9990	0.9951
Intraparticle diffusion	*k* _*i*1_	23.188	28.463	31.556	44.485	55.225
	*C*	−5.5100	−1.8826	−2.4937	−0.9429	10.448
	*R* _1_ ^2^	0.8930	0.9875	0.9877	0.9967	0.9446
	*k* _*i*2_	8.5084	5.5405	4.7350	2.1192	2.2246
	*C*	101.91	136.41	153.10	195.22	197.36
	*R* _2_ ^2^	0.9668	0.9819	0.9545	0.9806	0.3939

**Table 5 polymers-11-01412-t005:** Langmuir and Freundlich isotherm constants for adsorption of Pb(II) onto SPEN-CTAB-0%.

Isotherms	Parameters (Temperature = 298.15 K)	
Langmuir	*q*_m_ (mg/g)	281.69
	*K*_L_ (L/mg)	0.1117
	*R* _L_	0.08998
	*R* ^2^	0.9807
Freundlich	*K*_F_ ((mg/g) (L/mg)^1/*n*^)	12.861
	*n* ^−1^	0.1861
	*R* ^2^	0.8046

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
