# Peer review of "Functionalized Poly(arylene ether nitrile) Porous Membrane with High Pb(II) Adsorption Performance"

_polymers, 2019, doi:10.3390/polym11091412_

Round 1

Reviewer 1 Report

Current work describes the preparation of arylene ether nitrile porous membrane composite with CTAB, and its effect on Pb capture performance has been studied. The work is relevant can be published after addressing the following points.

1- While the effect of CTAB amount has been studied, the effect of CTAB is not clearly presented and hypothesis should be clarified.

2-Effect of pH has been studied however it is necessary to mention and discuss which functional groups should be protonated/deprotonated under what pH and effect of this to the overall Pb uptake should be discussed in the MS.

3- Porosity of these materials have been reported however it is not clear how the porosities are measured. If have not done so, N2 isotherm at 77K should be collected for BET and pore size distribution assignment and it should be included in the discussion.

4- In figure 3, authors show that with increasing CTAB amount, the contact angle is increasing, however, the water uptake also increases. How does this make sense?

5- Water vapor isotherms from a volumetric or gravimetric system would be helpful

6- It should be clearly stated in the intro the work has been done on the topic and how this study differs from others in literature and also it should be mentioned how does the Pb uptake capacity compare to others in the literature

Reviewer 2 Report

This paper is another one of a series of papers by the same research group concerning adsorbent formed from sulfonate poly(arlene ether nitrile) (SPEN).  As the authors described in the introduction part, heavy metal ion pollution has been one of serious problems and thus porous membranes which adsorb heavy metal ions have to be developed.  In this study, the authors prepared porous membranes from SPEN in the absence or presence of CTAB, and studied their adsorption behavior of Pb(II) ions.  I am wondering the molecular level structure of porous membranes, but I would like to recommend publication of this paper appropriate minor revision.  The followings are specific points.  

(1) 3.1. Characterization of SPEN.  The proton NMR spectrum for SPEN (Figure 1d) shows very sharp signals.  The authors should add the molecular weight and molecular weight distribution of the SPEN.  

(2) 3.2. Structure Characterization of Porous Membrane. 

(i) I am wondering the mechanism of the formation of porous membranes from SPEN.  The authors should indicate the reaction scheme of formation of porous materials from SPEN.  

(ii) I am interested in why CTAB can control the porosity of membranes.  

(iii) Nitrogen adsorption measurements may be also useful to characterize the porous membranes.  

(3) Figures 3, 4, 5, 6, 10, and 11 should indicate error bars.  

(4) There are some mistakes.  

Round 2

Reviewer 1 Report

Authors have revised the MS and addressed all the comments raised by reviewers. The paper can be published in the current form.